# Absorptive Turbulent Seawater and Parameter Optimization of Perfect Optical Vortex for Optical Communication

**Qingze Yan** [†] [iD] **, Yixin Zhang** [†]**, Lin Yu and Yun Zhu** *

School of Science, Jiangnan University, Wuxi 214122, China
* Correspondence:zhuyun1210@163.com
† They are the co-first authors.

**Abstract:** In this paper, the optimization of perfect optical vortex (POV) parameter for underwater wireless optical communication link under M-QAM by average bit-error rate (ABER) and the effect of seawtaer turbulence on link information capacity are investigated. The link is absorbent, weakly turbulent, and bandwidth-limited. In investigating, we use the spectral absorption coefficient to describe the wavelength effect of seawater absorption. Specifically, under the paraxial approximation and Rytov approximation conditions, we define the average signal-to-noise-crosstalk ratio including the system bandwidth factor and derive the bandwidth-limited ABER of the OAM carrier link. Capitalizing on the defined average signal-to-noise crosstalk ratio and the derived bandwidth-limited ABER of link, the novel closed-form expression for the average information capacity of the perfect optical vortex link under M-QAM modulation is proposed. Through the numerical analysis of the ABER and the average information capacity, the POV optimization parameters in specific communication links are obtained and new conclusions are drawn that the average information capacity is restricted by both signal wavelength and the seawater absorption coefficient.

**Keywords:** perfect optical vortex; absorptive turbulent seawater; parameter optimization; average information capacity; modulation





## 1. Introduction

Vortex beams with infinite N qubit orbital angular momentum (OAM) base [1–8] have attracted the attention of underwater wireless optical communication (WOC) researchers because of their strong resistance to environmental interference, high security, wide band width and low cost [8–12]. However, because the modes of OAM are determined by the wave-front spatial structure of the carrier, the wave-front distortion of vortex beams caused by ocean turbulence will inevitably lead to crosstalk among OAM modes [8–14]. To this end, the attempt to use vortex beams such as Gauss-Schell model [8], Bessel-Gauss [9], Hermite-Gauss [10], Lommel [13] and local wave [14] as signal sources to reduce the turbulent sea effect of OAM mode transmission and improve the transmission quality of underwater OAM mode has become one of the most popular research topics in this field.

In 2013, Ostrovsky et al. put forward perfect optical vortex (POV) new beam with dark hollow radius independent of topological charge, and approximately realized this beam in Fourier transform optical experiments using computer-controlled liquid crystal spatial light modulation [15]. After, Pravin Vaity et al. deduced the mathematical model of POVs by Fourier transform of the Bessel beam, and verified the mathematical model of POVs by Fourier transform of the Bessel beam experimentally. At present, this model has become a general model for theoretical research on POVs characteristics [16]. POVs have the characteristic of having the same initial ring width and radius under different topological charges, and in the source plane, there is no sidelobe of intensity distribution. These new characteristics of POV have been studied extensively through the evolution of transmission media such as no turbulence free space [17,18], turbulent atmosphere [19],

and underwater [20–23]. These studies show that: The POV carrying OAM can be used as communication link information carrier by OFDM 16-QAM modulation [17]. In 2-channel OAM multiplexed WOC link, POVs carrier links perform better than Laguerr-Gaussian (LG) carrier links [18]. In a turbulent atmosphere experiment, the POV beam retains its original characteristics and its propagation distance depends on the power of the light source, the parameters of the Fourier objective, and the model of spatial light modulator [19]. POV transmission experiments under different underwater conditions show that the bit-error rate (BER) of OAMs is higher than its ideal value at lower SNR, even in stagnant water, and the bit-error rate reaches its maximum value when the beam propagates in bubble underwater links [20]. In an anisotropic oceanic turbulence, the self-focusing property of POVs improves to the transmission quality of OAM modes [21]. Although the wavelength, topological charge and radius-thickness ratio of POVs hardly affect the propagation quality of POV beam in ocean turbulence, the radius of POV beam does [22]. In the short link of absorbent turbulent seawater, POVs have a long wavelength within the "seawater window wavelength" interval, a small topological charge of OAM, a large annular radius, and a small aperture of transmitter and receiver; the probability of receiving POV is high [23]. Recently, the underwater propagation properties of PLG, a new beam similar to POV, have also received attention. In the case of selected beam types, selecting appropriate system parameters is also one of the important methods to effectively control the impact of ocean turbulence [24], but the parameter optimization of existing POVs has not been studied [15–23]. However, as far as we know, the optimization of source parameters of UWOC link with M-QAM and POV signal carrier, and the influence of wavelength dependence of seawater absorption on the average information capacity (AIC) of POV carrier link have not been reported.

The main contributions of this article are mainly summarized as follows: combined with the definition of signal-to-noise ratio(SNR) for bandwidth-constrained systems and the discussion of zero mean additive Gaussian channel, we defined a new average signal-crosstalk-noise ratio (ASCNR) of OAM channel that is connected to the signal bandwidth. The AIC and ABER of bandwidth-constrained link with signal modulation of M-QAM has been derived in a closed form. For a given bandwidth-constrained link, we find that choosing a suitable modulation order M can reduce the bandwidth limited ABER caused by the channel of turbulent absorption seawater.

Seawater scattering, bubbles, physio-chemical properties of seawater, and pointing errors are other important factors affecting optical communication in seawater, but seawater scattering, bubbles and physio-chemical properties of seawater are very complex factors affecting optical communication transmission [25–29], so we only consider a POV in clean or ignore the complicated factors of optical communication problems in offshore underwater channels. Therefore, the optimization of source parameters and AIC of underwater wireless optical communication absorption link with carrier of POV and M-QAM are investigated under Rytov approximation. In Section 2, normalized amplitude of POV in weak absorptive turbulence is presented. In Section 3, the bandwidth limited ABER of POV carrier link using M-QAM is derived. In Section 4, the AIC of the POV carrier and bandwidth-constrained link using M-QAM is proposed. Based on the numerical simulation of bandwidth-limited ABER of M-QAM link and the AIC of the link, the optimized POV source parameters and the influence of seawater turbulence and absorption on the AIC of the communication link are given in Section 5, respectively. Finally, conclusions are given in Section 6.

## 2. Normalized Amplitude Perfect Optical Vortex in Weak Turbulent and Absorptive Seawater

Figure 1 depicts a schematic representation of an underwater wireless optical communication system model.

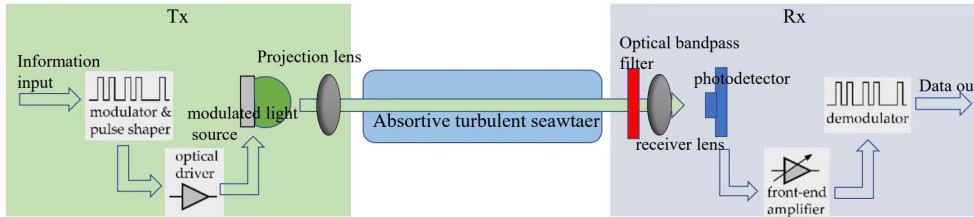

**Figure 1.** Schematic of underwater wireless optical communication system model.

In this model, the transmitter (Tx) is composed of M-QAM modulator, optical driver, modulated light source and projection lens. The receiver (Rx) is made of an optical band-pass filter, receiver lens, photodetector, front-end amplifier and demodulator. The light POV is studied in weakly scintillated turbulent seawater, $\sigma_{1\epsilon\chi_T}^2 = 1.23C_{\epsilon\chi_T}^2 k^{7/6}z^{11/6} < 1$ [30]. Paraxial transmission and Rytov approximations are adopted [31]. In cylindrical coordinates $(\rho, \theta, z)$ and turbulent absorbent seawater, the normalized complex amplitude of POV propagation is expressed as [23]

$$
E_{om}(\rho,\theta,z) = \frac{\mathrm{i}^{m_0}}{w(z)}\sqrt{\frac{2\exp(\frac{r_0^2}{\tilde{w}^2})}{\pi\left|\mathrm{I}_{m_0}(\frac{r_0^2}{\tilde{w}^2})\right|}}\mathrm{J}_{m_0}\left[\frac{2\rho r_0}{\tilde{w}w(z)}\exp(\mathrm{i}\Theta)\right]\exp\left[\mathrm{i}m_0\theta + \mathrm{i}\Theta + \frac{\mathrm{i}kn_R z}{2R(z)}(\rho^2+r_0^2)\right] \quad (1)
$$

$$
\times\exp\left[-k_0 n_I z\left(1 + \frac{\rho^2+r_0^2}{2R(z)}\right) + \mathrm{i}k_0 n_R z\right]\exp\left[-\frac{\rho^2 - (zr_0/z_{oR})^2}{w(z)^2} + \Psi(\rho,\theta,z)\right],
$$

where $w(z) = \sqrt{1 + z^2/z_{oR}^2}$, $\tilde{w}$ represents the half beam-width of the POV at the image plane of the Fourier lens, $z_{oR} = k(n_R + \mathrm{i}n_I)\tilde{w}^2/2$ is the Rayleigh range, $k_0$ is the wave number of light transmitted in a vacuum , $n_R$ and $n_I$ are the real part and the imaginary part of the $n_{oc}$, respectively; $m_0$ is the OAM topological charge of initial OAM signal, $r_0$ is the ring radius of POV at the object plane of the Fourier lens, $\mathrm{J}_{m_0}$ is the $m_0$ th-order first kind Bessel function, $\Theta = arctan(z/z_R)$, $R(z) = z + z_{oR}^2/z$, and $\Psi$ is the complex phase perturbations caused by seawater turbulence. $n_I = \lambda c_{oc}(\lambda)/(4\pi)$ [32], and $c_{oc}(\lambda)$ is the spectrum absorption coefficient of seawater.

### 3. Average Bit-Error Rate of Link

By Fourier's orthogonal series expansion theory, we can expand the complex amplitudes expressed in Equation (1) into series based on orthogonal plane helical basis $\exp(\mathrm{i}m\theta)$ [14]

$$
E_m(\rho,\theta,z) = \sum_m \alpha_m(\rho,z)\exp(\mathrm{i}m\theta), \quad (2)
$$

where $m$ is a new OAM topological charge of the random POV in turbulent seawater.

The expansion coefficient $\alpha_m(\rho,z)$ in Equation (2) is a random function of position and distance and is given by

$$
\alpha_m(\rho,z) = \frac{1}{2\pi}\int_0^{2\pi} E_m(\rho,\theta,z)\exp(-\mathrm{i}m\theta)d\theta. \quad (3)
$$

The probability distribution function of receiving OAM mode is obtained by $p(m/m_0) = \langle\alpha_m(\rho,z)\alpha_m^*(\rho,z)\rangle$, where $\langle\cdot\rangle$ represents the mean of turbulence ensemble, and its mathematical expression is

$$
p(m/m_0) = \frac{1}{4\pi^2}\int_0^{2\pi}\int_0^{2\pi} E_m(\rho,\theta,z)E_m^*(\rho,\theta',z)\exp[-\mathrm{i}m(\theta-\theta')d\theta d\theta']. \quad (4)
$$

For stable stratification and isotropic turbulence, and taking into account the integral [33]

$$\int_0^{2\pi} \exp[\tau \cos(\theta - \phi) - im\theta]d\theta = 2\pi I_m(\tau)\exp(-im\phi), \tag{5}$$

and applying Equation (1) in (4), we have the conditional probability distribution of the OAM mode carried by the POV [23]

$$p(m/m_0) = \frac{2\exp\left(\frac{r_0^2}{\bar{w}^2} - 2k_0 n_I z\left(1 + \frac{r_0^2}{2R(z)}\right)\right)}{\pi w(z)^2 \left|I\left(\frac{r_0^2}{\bar{w}^2}\right)\right|} \exp\left[\frac{8(zr_0)^2 n_R^2}{w(z)^2 k_0^2 \bar{w}^4 (n_R^2 + n_I^2)}\right] \tag{6}$$

$$\times \exp\left(-\frac{\rho^2}{w_{eff}^2}\right) \left|J_{m_0}\left[\frac{2\rho r_0}{\bar{w}w(z)}\right]\right|^2 I_{m-m_0}\left[\frac{2\rho^2}{\rho_0^2}\right],$$

where $I_{m_0}(x)$ is the modified Bessel function of the first kind and $m_0$-th orders.

In Equation (6) $\rho_0$ is the transverse spatial coherence radius of the plane wave and is given by

$$\rho_0 = \left\{ \frac{9.444 C_{\epsilon\chi_T}^2 (n_r^2 + n_I^2)z}{\lambda^2(1 - \overline{\omega})^2} \left[ \overline{\omega}^2 \kappa_0^{1/3} U(2; \frac{7}{6}; \frac{\kappa_0^2 \eta^2}{R_T^2}) + \overline{\omega}^2 4.6 \eta^{2/3} \kappa_0 \Gamma(\frac{7}{3}) U(\frac{7}{3}; \frac{3}{2}; \frac{\kappa_0^2 \eta^2}{R_T^2}) \right. \right.$$

$$\times \kappa_0^{1/3} U(2; \frac{7}{6}; \frac{\kappa_0^2 \eta^2}{R_S^2}) + 4.6 \eta^{2/3} \kappa_0 \Gamma(\frac{7}{3}) U(\frac{7}{3}; \frac{3}{2}; \frac{\kappa_0^2 \eta^2}{R_S^2}) - 2\overline{\omega}\kappa_0^{1/3} U(2; \frac{7}{6}; \frac{\kappa_0^2 \eta^2}{R_{TS}^2}) \tag{7}$$

$$\left. \left. - 9.2 \eta^{2/3} \kappa_0 \Gamma(\frac{7}{3}) U(\frac{7}{3}; \frac{3}{2}; \frac{\kappa_0^2 \eta^2}{R_{TS}^2}) \right] \right\},$$

where $C_{\epsilon\chi_T}^2 = 0.809 \times 10^{-7} \epsilon^{-1/3} \chi_T \overline{\omega}^{-2}(1 - \overline{\omega})^2 \left(K^2/m^{2/3}\right)$ is the structural constant of the salinity-temperature fluctuation of oceanic turbulence that is a function of the rate of dissipation of kinetic energy per unit mass of fluid $\epsilon$ in the range $[10^{-10}, 10^{-1}]m^2/s^3$ the dissipation rate of the mean-squared temperature $\chi_T$ in the range $[10^{-10}, 10^{-2}]K^2/s$ and $\overline{\omega}$ is the ratio of temperature and salinity contributions to the refractive index fluctuations, which can vary in the interval $(-5, 0)$. Here $\overline{\omega} = -5$ temperature fluctuation is dominant in the induction of optical turbulence, while $\overline{\omega} = 0$ salinity fluctuation is dominant in the induction of optical turbulence. $\kappa$ is the scalar spatial wave number of turbulent fluctuation, $\kappa_0 = 2\pi/L_0$, $\kappa_T = R_T/\eta$, $\kappa_S = R_S/\eta$, $\kappa_{TS} = R_{TS}/\eta$, $\eta$ is turbulent inner scale, $L_0$ is the outer scale of turbulence, $R_j = \sqrt{3}\left[W_j - 1/3 + 1/(9W_j)\right]^{3/2}/Q^{-3/2}(j = T, S, TS)$, $W_j = \left\{ \left[\frac{Pr_j^2 Q^4}{(6\beta)^2} - \frac{Pr_j Q^2}{81\beta}\right]^{1/2} - \left[\frac{1}{27} - \frac{Pr_j Q^2}{6\beta}\right] \right\}^{1/3}$, $Q$ is the non-dimensional constant; $Pr_T$ and $Pr_S$ respectively represent the Prandtl number of the temperature and salinity, $Pr_{TS} = 2 Pr_T Pr_S /(Pr_T + Pr_S)$; $w_{eff} = \left(\frac{2}{w^2(z)} + \frac{2}{\rho_o^2} + \frac{k_0 n_I z}{R(z)}\right)^{-1/2}$ is defined as the effective radius of the POV in the absorbed seawater turbulence.

Note the structural constants $C_{\epsilon\chi_T}^2$ of the salinity-temperature fluctuation of oceanic turbulence increases with the increase of the salinity fluctuation, but it decreases with the increase of the temperature fluctuation.

The received probability of OAM modes of POV can be obtained

$$P(m \mid m_0) = 2\pi \int_0^{D/2} p(m \mid m_0)\rho d\rho, \tag{8}$$

where $D$ is the receiver diameter. For $m \neq m_0$, $P(m \mid m_0)_{m \neq m_0}$ is the received probability of OAM crosstalk modes that describes the probability of the photon migrating from the

OAM signal mode to the new OAM mode, and $m = m_0$, $P(m \mid m_0)_{m=m_0}$ is the received probability of OAM signal modes.

It has been shown in [34] that OAM crosstalk distribution follows the zero-mean Gaussian distribution. Therefore, the communication link studied in this paper is still a zero mean additive Gaussian channel. Combined with the definition of SNR for bandwidth-constrained systems in [35] and the discussion of zero mean additive Gaussian channel in [30], we define a new average signal-crosstalk-noise ratio (ASCNR) of OAM channel as

$$\gamma_{m,m_0} = \frac{P^2(m/m_0)_{m=m_0}}{[\sum_{m=-\infty}^{\infty} P(m/m_0) + N_0/P_{TX}]2B}, \tag{9}$$

where $P_{TX}$ is the transmit power and $N_0$ is the transmitter and receiver noise power, $B = 2R_b/\log_2 M$ is the bandwidth occupied by modulated signal, and $R_b$ is the bit rate [36].

The Equation (9) can also be rewritten in terms of the bit rate

$$\gamma_{m,m_0} = \frac{P^2(m/m_0)_{m=m_0}\log_2 M}{[\sum_{m=-\infty}^{\infty} P(m/m_0) + N_0/P_{TX}]4R_b}. \tag{10}$$

Equation (10) indicates that when other system parameters remain unchanged, to maintain the ASCNR of the transmitted OAM signal, when the length of the communication channel or the turbulence strength within the channel increases, we must reduce the communication bandwidth or bit rate or increase the modulation order $M$.

For signal modulation of M-QAM, based on (9) [37], we can obtain the bandwidth-limited ABER of the OAM $m$th channel carried by the POV as

$$P_B = \frac{2}{\log_2 M}\left(1 - \frac{1}{\sqrt{M}}\right) \mathrm{erfc}\sqrt{\frac{3\log_2 M}{2(M-1)}\gamma_{m,m_0}}$$

$$\times \left[1 - \frac{1}{2}\left(1 - \frac{1}{\sqrt{M}}\right)\mathrm{erfc}\sqrt{\frac{3\log_2 M}{2(M-1)}\gamma_{m,m_0}}\right], \tag{11}$$

where $\mathrm{erfc}(\cdot)$ denotes the complementary error function.

## 4. Average Information Capacity of Link

Figure 2 shows the discrete memoryless channel among the energy levels of signal OAM and received OAM, which is characterized by channel (transition) probabilities. Let $X = \{m_{00}, m_{01} \cdots, m_{0i-1}\}$ and $Y = \{m_0, m_1 \cdots, m_{J-1}\}$ be the input and the output letters channel, respectively; the channel is completely characterized by the following set of transition probabilities [36]:

$$p(m_j/m_{oi}) = P(Y = m_j/X = m_{0i}), 0 \le p(m_j/m_{0i}) \le 1, \tag{12}$$

where $i \in \{0, 1, \cdots, I-1\}$, $j \in \{0, 1, \cdots, J-1\}$, while $I$ and $J$ denote the sizes of input and output alphabets, respectively. The transition probability $p(m_j \mid m_{0i})$ represents the conditional probability that $Y = m_j$ for given input $X = m_{0i}$.

The information capacity is obtained by maximization of mutual information over all possible input distributions [36]:

$$C = \max_{\{p(m_{0i})\}} I(X, Y) = \max_{\{p(m_{0i})\}} [H(X) - H(X \mid Y)], \tag{13}$$

where $H(X) - \langle\log_2 P(X)\rangle$ represents the uncertainty about the channel input before observing the channel output, while $H(X \mid Y)$ represents the uncertainty of the channel input after it has been received.

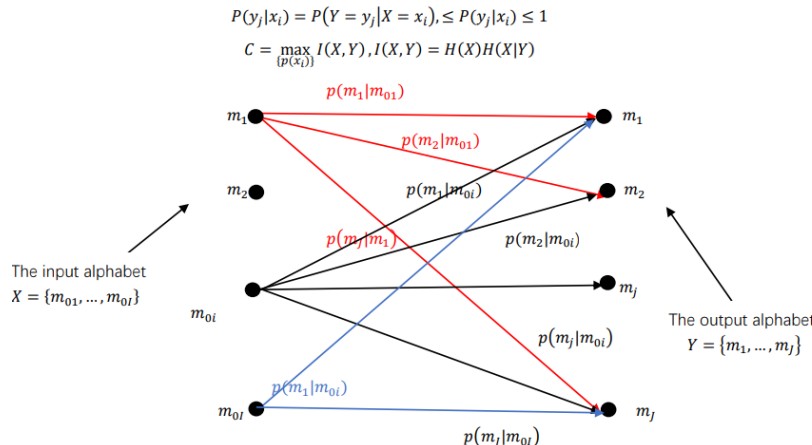

$$P(y_j|x_i) = P(Y = y_j|X = x_i), \leq P(y_j|x_i) \leq 1$$

$$C = \max_{\{p(x_i)\}} I(X,Y), I(X,Y) = H(X)H(X|Y)$$

**Figure 2.** Discrete memoryless channel among the energy levels of signal OAM and received OAM.

In Equation (13), $I(X;Y)$ is defined as mutual information. The mathematical expression for mutual information is expressed as follows:

$$
\begin{aligned}
I(X;Y) &= H(X) - H(X \mid Y) \\
&= \sum_{i=1}^{M} p(m_{0i}) \log_2\left[\frac{1}{p(m_{0i})}\right] - \sum_{j=1}^{N} p(m_j) \sum_{i=1}^{M} p(m_{0i} \mid m_j) \log_2\left[\frac{1}{p(m_{0i} \mid m_j)}\right].
\end{aligned}
\tag{14}
$$

Equation (14) represents that the mutual information is obtained as the output information minus information lost in the channel.

Considering that (1) POV constitutes an infinite OAM channel and the channels used for communication are finite: (2) when POV is transmitted in weakly turbulent seawater, the difference between the energy level corresponding to OAM crosstalk formed by seawater turbulence disturbance and the energy level of OAM signal is very small. Therefore, the OAM channel composed of $N = 2m + 1$ input and $N = 2m + 1$ output is a symmetric channel (MSC). Further, let constellation points be marked with gray codes, and the symbol error causes the corruption of only 1 bit out of $\log 2M$ bit comprising the symbol [38]. We have that $p(m_j \mid m_{0i}) = P_B \log_2 M/(M-1)$ and $p(m_j \mid m_{0j}) = 1 - P_B \log_2 M$, where $P_B \log_2 M$ is symbol error probability, $P_B$ is the bit-error rate (ABER).

The AIC of a bandwidth-constrained link is described by the following formula

$$
C = \log_2 N + (1 - P_B \cdot \log_2 M) \cdot \log_2(1 - P_B \cdot \log_2 M) + P_B \cdot \log_2 M \cdot \log_2 \frac{P_B \cdot \log_2 M}{N-1}.
\tag{15}
$$

Taking into account Equations (11) and (15), we can obtain the AIC of the POV carrier link with M-QAM as

$$
\begin{aligned}
C = \log_2 N &+ \left\{1 - 2\left(1 - \frac{1}{\sqrt{M}}\right) \operatorname{erfc}\sqrt{\frac{3\log_2 M}{2(M-1)}\gamma_{m,m_0}}\left[1 - \frac{1}{2}\left(1 - \frac{1}{\sqrt{M}}\right) \operatorname{erfc}\sqrt{\frac{3\log_2 M}{2(M-1)}\gamma_{m,m_0}}\right]\right\} \\
\times \log_2 &\left\{1 - 2\left(1 - \frac{1}{\sqrt{M}}\right) \operatorname{erfc}\sqrt{\frac{3\log_2 M}{2(M-1)}\gamma_{m,m_0}}\left[1 - \frac{1}{2}\left(1 - \frac{1}{\sqrt{M}}\right) \operatorname{erfc}\sqrt{\frac{3\log_2 M}{2(M-1)}\gamma_{m,m_0}}\right]\right\} \\
+ &\left\{1 - 2\left(1 - \frac{1}{\sqrt{M}}\right) \operatorname{erfc}\sqrt{\frac{3\log_2 M}{2(M-1)}\gamma_{m,m_0}}\left[1 - \frac{1}{2}\left(1 - \frac{1}{\sqrt{M}}\right) \operatorname{erfc}\sqrt{\frac{3\log_2 M}{2(M-1)}\gamma_{m,m_0}}\right]\right\} \\
\times \log_2 \frac{2}{N-1}&\left\{1 - 2\left(1 - \frac{1}{\sqrt{M}}\right) \operatorname{erfc}\sqrt{\frac{3\log_2 M}{2(M-1)}\gamma_{m,m_0}}\left[1 - \frac{1}{2}\left(1 - \frac{1}{\sqrt{M}}\right) \operatorname{erfc}\sqrt{\frac{3\log_2 M}{2(M-1)}\gamma_{m,m_0}}\right]\right\},
\end{aligned}
\tag{16}
$$

where $N = -m, -m+1 \cdots, 0, \cdots m-1, m$, and AIC represents the maximum data rate that the system can achieve.

## 5. Numerical Analysis

In this section, we will search for the optimal light source parameters of M-QAM modulated POV communication links in the central Pacific [39] under weak turbulence through numerical simulation of bandwidth-limited ABER, and discuss the influence of seawater turbulence and absorption on the AIC of the POV carrier link with M-QAM. For the sake of argument, unless otherwise specified, we choose the parameters of the absorbable turbulent seawater link as Table 1, $\tilde{w} = \tilde{w}_{opt}(z = z_0)$ and we adopted the current reachable experimental distance of $z = 150$ m [40,41] and modulation order $M = 256$ as the link length and modulation order [40]. Note that the maximum transmission distance achieved in the existing experiments is 200 m [42], but the turbulence effect in seawater is not taken into account. Considering that seawater turbulence will cause crosstalk loss of the OAM signal, and the wider the bandwidth, the greater the crosstalk loss of the OAM signal, we adopt a narrower bandwidth ($B = 0.5$ GHZ) and a shorter distance ($z = 150$ m) in numerical calculation.

**Table 1.** Major system parameters.

| Symbol | Parameters | Value |
|---|---|---|
| $\mathrm{Pr}_T$ | Prandtl number of the temperature | 0.72 |
| $\mathrm{Pr}_S$ | Prandtl number of the salinity | 700 |
| $Q$ | Non-dimensional constant | 2.5 |
| $B$ | Bandwidth | 0.05 GHZ |
| $C_{\varepsilon\chi_T}^2$ | Turbulence strength | $10^{-13}$ K$^2$/m$^{2/3}$ |
| $\varepsilon$ | Rate of dissipation of kinetic energy per unit mass of fluid | $10^{-3}$ m$^2$/s$^3$ |
| $\chi_T$ | Dissipation rate of the mean-squared temperature | $10^{-7}$ K$^2$/s |
| $\overline{\omega}$ | Ratio of temperature and salinity | $-4.5$ |
| $k_0$ | Wave number | $2\pi/\lambda$ |
| $w_{eff}$ | Effective radius | $\left(\frac{2}{w^2(z)} + \frac{2}{\rho_o^2} + \frac{k_0 n_I z}{R(z)}\right)^{-1/2}$ |
| $w(z)$ | Half beam-width | $\sqrt{1 + z^2/z_{oR}^2}$ |
| $r_0$ | Ring radius | 0.03 m |
| $\eta$ | Inner scale | 0.01 m |
| $L_0$ | Outer scale | 10 m |
| $m_0$ | OAM topological charge | 1 |
| $D$ | Receiver diameter | 0.05 m |
| $\lambda$ | Wavelength | 470 nm |
| $n_I$ | Imaginary part of the $n_{oc}$ | $0.4487 \times 10^{-9}$ |
| $z$ | Propagation distance | 150 m |

### 5.1. Probability Distribution and Received Probability

Figure 3 shows the transformation of received probability of the OAM signal and crosstalk modes carried by POVs with wavelength $\lambda = 470$ nm ($n_I = 0.4487 \times 10^{-9}$) at transmission $z = 150$ m in seawater. In Figure 3a the diameter of receiver is 0.02 m, while in Figure 3b the diameter of is 0.005 m. By comparing the bar charts in Figure 3a,b, it can be found that (1) the optical communication link adopting the receiver of small aperture will cause serious loss of OAM signal owning high order topological charge, and it will cause a large loss of all OAM signals; (2) the receiving probability of the OAM signal is independent of topological charge when the communication link adopts a larger aperture to receive the OAM signal. This result is very beneficial to the mode division multiplexing of communication link and is the purpose of designing the POV. The curves in Figure 3a show that even though the channel has reached its experimental length 150 m for $D = 0.02$ m, OAM crosstalk produced by seawater turbulence is still very small. This result indicates that POVs transmitted in turbulent absorbing seawater still have the characteristics of their

topological charges transmitted in free space [15,16], that is, intensity is independent of their topological charges.

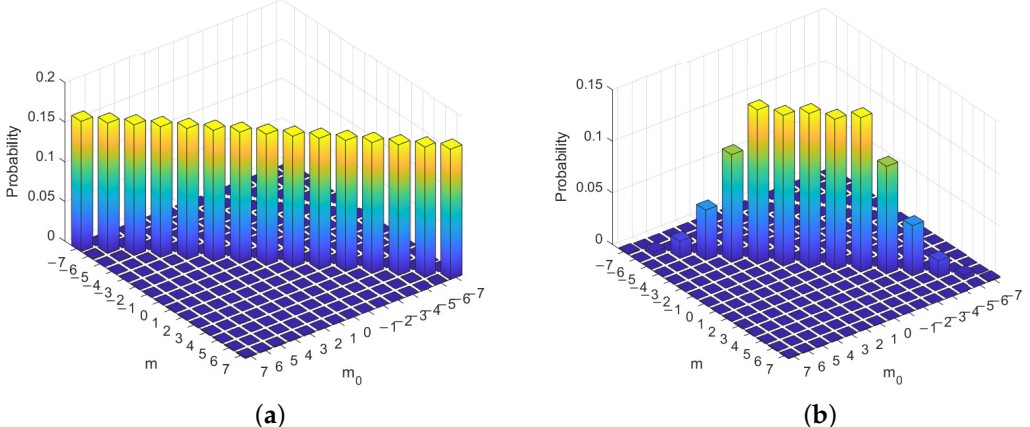

(a)        (b)

**Figure 3.** The received probability of OAM signal modes carried by POV versus different topological charge of initial OAM modes and topological charge of new OAM modes for $r_0 = 0.03$ m, $\overline{\omega} = -4.5$, $n_I = 0.4487 \times 10^{-9}$. (**a**) $z = 150$ m, $D = 0.02$ m; (**b**) $z = 150$ m, $D = 0.005$ m.

### 5.2. The Bandwidth, Modulation Order, SNR of Optical System and ABER

In this sub-section, we will analyze the influence of modulation order $M$, the bandwidth and link channel on ABER.

To analyze which of M-QAM is more advantageous for the underwater POV transmission channel, in Figure 4 we give the numerical results for the bandwidth-limited ABER of POV carrier link with M-QAM versus the propagation distance and turbulence strength for $B = 0.05$ GHz. The Figure 4 shows the bandwidth-limited ABER of link increases with the increase of propagation distance $z$ for each modulation order $M$. Further, in $C^2_{\varepsilon\chi_T} = 10^{-13}$ K$^2$/m$^{2/3}$, for $z \leq 123$ m, ABER of 4-QAM is the lowest, but for $z \geq 123$ m, ABER of 512-QAM is the lowest. In $C^2_{\varepsilon\chi_T} = 10^{-15}$ K$^2$/m$^{2/3}$ (see Figure 4a), ABER of 4-QAM is always the lowest. That is, low order modulation has better performance over shorter distances, while high order modulation has advantages over long distances.

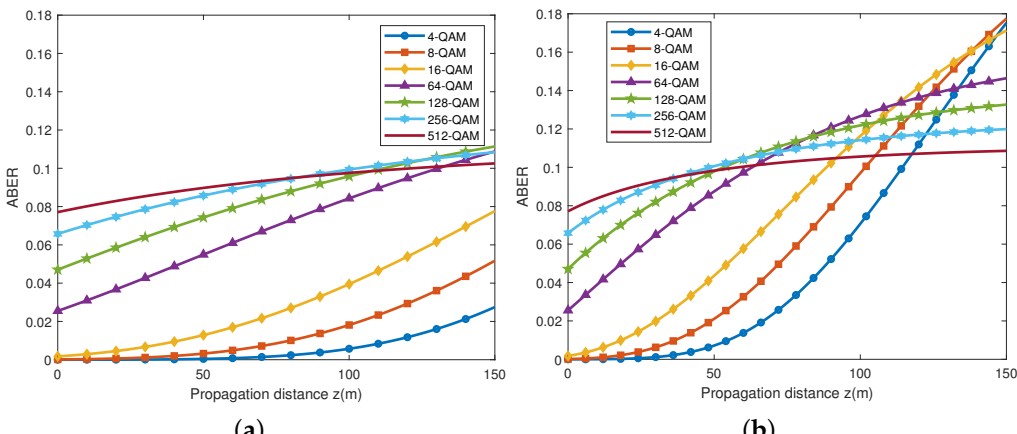

(a)        (b)

**Figure 4.** The bandwidth-limited ABER of the POV carrier link with M-QAM versus the propagation distance in the turbulence strength of (**a**) $C^2_{\varepsilon\chi_T} = 10^{-15}$ K$^2$/m$^{2/3}$, (**b**) $C^2_{\varepsilon\chi_T} = 10^{-13}$ K$^2$/m$^{2/3}$.

Figure 5 show the evolutive curves of bandwidth-limited ABER of link with variation of the M-QAM and the turbulence strength. It can be seen from Figure 5 that the bandwidth-limited ABER increases with the increase of turbulence strength for M-QAM. Figure 5a shows that in $z = 100$ m, from $C^2_{\varepsilon\chi_T} = 10^{-15}$ K$^2$/m$^{2/3}$ to $10^{-13}$ K$^2$/m$^{2/3}$, the bandwidth-limited ABER of 4-QAM is always the lowest. While Figure 5b shows that, in

$z = 150$ m, as $C^2_{\varepsilon\chi_T} \geq 10^{-14}$ K$^2$/m$^{2/3}$ , when $C^2_{\varepsilon\chi_T}$ continues to increase, the modulation order M of M-QAM corresponding to the minimum value of bandwidth-limited ABER increases successively.

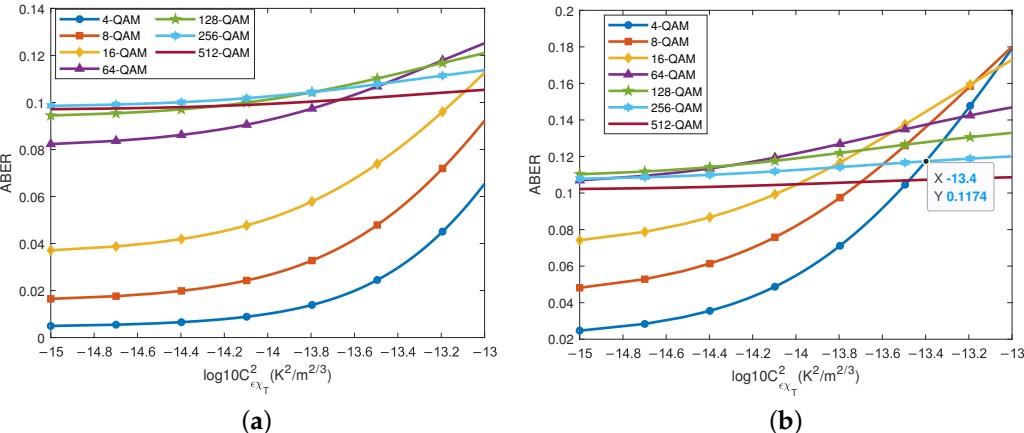

(**a**)          (**b**)

**Figure 5.** The bandwidth-limited ABER of POV carrier link with M-QAM versus turbulence strength in the link length of (**a**) $z = 100$ m, (**b**) $z = 150$ m.

More importantly, both Figures 4 and 5 indicate that, in large turbulence fluctuation $\sigma^2_{1\varepsilon\chi_T} = 1.23 C^2_{\varepsilon\chi_T} k^{7/6} z^{11/6}$ , high-order QAM has better anti-interference ability than low-order QAM.

In Figure 6 we give the numerical results for the bandwidth-limited ABER of POV carrier link with M-QAM versus the bandwidth $B$ for different propagation distance and turbulence strength. Figure 6 displays that the bandwidth limited ABER of link increases with the increase of bandwidth $B$, this is obviously because the increase of lead to the decrease of SCNR (see Equation (9)). Specifically, comparing the change in propagation distance between Figure 6a,b and the change in turbulence strength between Figure 6b,c, we find that for the short length of communication link with narrow communication bandwidth in weak seawater turbulence, QAM of low order $M$ should be adopted. However, for long communication links with wide communication bandwidth in strong seawater turbulence, QAM of high order $M$ should be adopted.

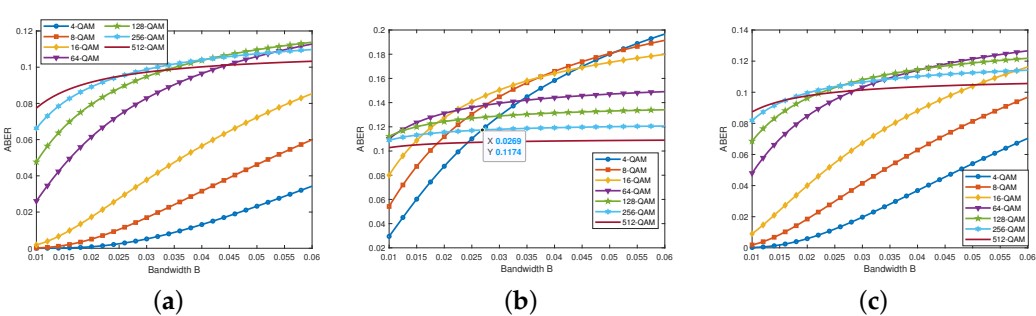

(**a**)          (**b**)          (**c**)

**Figure 6.** The bandwidth-limited ABER of POV carrier link with M-QAM versus (**a**) the propagation distance (**b**) turbulence strength. (**a**) $z = 75$ m, $C^2_{\varepsilon\chi_T} = 10^{-13}$ K$^2$/m$^{2/3}$, (**b**) $z = 150$ m, $C^2_{\varepsilon\chi_T} = 10^{-13}$ K$^2$/m$^{2/3}$, (**c**) $z = 150$ m, $C^2_{\varepsilon\chi_T} = 10^{-14}$ K$^2$/m$^{2/3}$.

In Figure 7, we give the numerical results for the bandwidth-limited ABER of the POV carrier link with M-QAM versus the SNR of optical system $P_{TX}/N_0$. Figure 7 displays that the bandwidth limited ABER of the link decreases with the increase of the $P_{TX}/N_0$, that is, the increase of SNR $P_{TX}/N_0$ of optical system contributes to the ability of the communication system to resist turbulent channel disturbance. This is obviously because the decrease of $P_{TX}/N_0$ leads to the decrease of SCNR (see Equation (9)). Figure 7 also shows that for modulation order $M \leq 16$ and the SNR $P_{TX}/N_0 < 25$ dB, the ABER of the

link decreases with the increase of SNR $P_{TX}/N_0$, but when $P_{TX}/N_0 > 25$ dB, the increase of $P_{TX}/N_0$ does not improve the link ABER performance. For modulation order $M > 16$, the effect of improving link ABER performance by increasing $P_{TX}/N_0$ is no longer obvious. The reason for this result is that the total noise in ASCNR is the sum of the noise components of the optical system and the turbulence-induced OAM crosstalk noise components (see Equation (9)). When the $P_{TX}$ is fixed, the noise of the optical system decreases with the increase of $P_{TX}/N_0$, thus the contribution of the noise component of the optical system to the total noise decreases, the turbulence-induced OAM crosstalk noise is the dominant noise. Further from the relationship between ABER and the ASCNR (Equation (11)), it is not difficult to understand the numerical results of this graph.

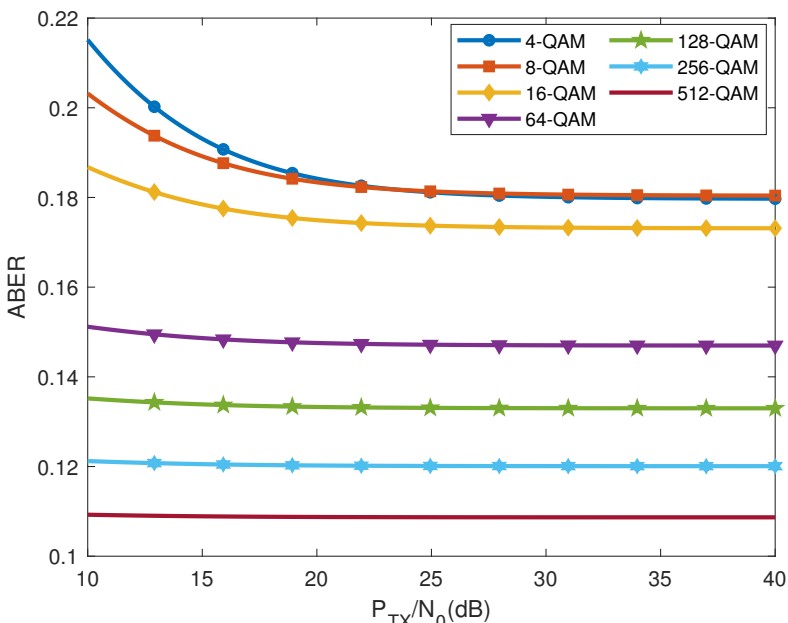

**Figure 7.** The bandwidth limited ABER of POV carrier link with M-QAM versus the SNR of optical system $P_{TX}/N_0$.

*5.3. System Parameter Optimization*

In this sub-section, we will analyze the influence of link channel on ABER to get the optimal signal parameters of communication link.

Figures 8 show the variation of bandwidth-limited ABER of the POV carrier link using 4-QAM/256-QAM with a half beam-width of POV, topological charge $m_0$ of initial signal modes and channel length $z$. In Figure 8a we draw curves of bandwidth-limited ABER of POV communication link as a function of the half beam-width of the POV at the ends of four different channel lengths. Figure 8a,c show that there exists optimal with minimum bandwidth-limited ABER as the function of the channel length (see Table 2). Figure 8b,d indicate that $\tilde{w}_{opt}$ is related to the OAM topological charge $m_0$ of initial signal modes (see Table 3). Combining the conclusions drawn from Figure 8a–d, we conclude that optimization $\tilde{w}$ is a function of channel length $z$ and the OAM topological charge of signal modes, that is $\tilde{w}_{opt}$. In the subsequent discussion of this paper, we will replace half beam-width with the optimization half beam-width.

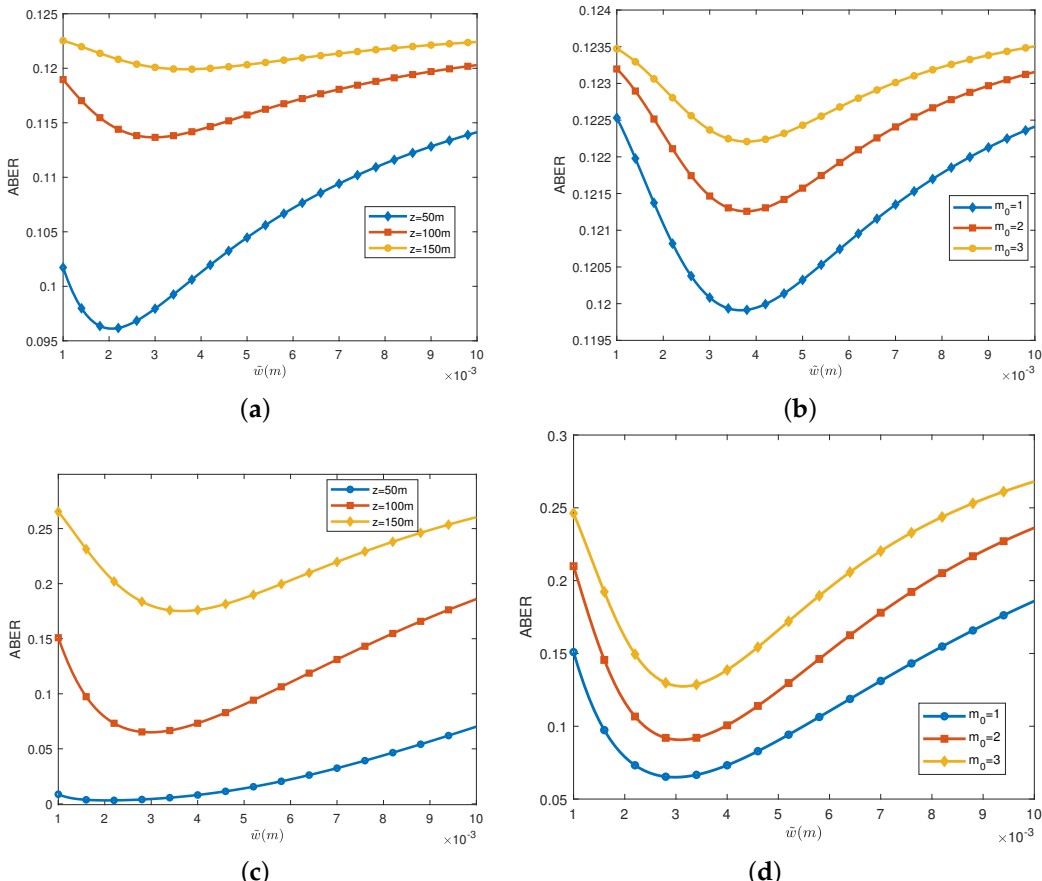

**Figure 8.** The bandwidth-limited ABER of POV carrier link versus the half beam-width $\tilde{w}$ and topological charge $m_0$ under modulation of (**a**) 256-QAM, (**b**) 256-QAM, (**c**) 4-QAM and (**d**) 4-QAM.

**Table 2.** The Relationship between the Optimal Half-width of the POV and the Channel Length.

| Modulation System | Length of the Channel $z$ (m) | Optimum Half Width $\tilde{w}_{opt}$ (m) |
|---|---|---|
| 4-QAM/256-QAM | 50 | 0.0021 |
| 4-QAM/256-QAM | 100 | 0.0032 |
| 4-QAM/256-QAM | 150 | 0.0038 |

**Table 3.** The Relationship between the Optimal Half Beam-width of POV and the Topological Charge of OAM.

| Modulation System | OAM Topological Charge $m_0$ | Optimum Half Width $\tilde{w}_{opt}$ (m) |
|---|---|---|
| 4-QAM/256-QAM | 1 | 0.0036 |
| 4-QAM/256-QAM | 2 | 0.0037 |
| 4-QAM/256-QAM | 3 | 0.0038 |

*5.4. Channel Capacity of the Link*

In this sub-section, we apply the conclusions obtained in the previous section, select the optimized source parameter value, and select the 4-QAM/256-QAM modulation schemes to discuss the influence of various factors in weak turbulence absorption seawater on the average information capacity of OAM link.

We know that the turbulence fluctuation of long wavelength OAM signal is lower than that of short wavelength signal [31] for a given absorption coefficient, and the loss of OAM signal transmitted in long absorption seawater channel is higher than that in short channel

for a given signal wavelength. However, the absorption coefficient of seawater depends on the wavelength of light [39]. Therefore, seawater absorption as well as signal wavelength are two important factors for the performance research of optical communication system with OAM mode carrier.

In Figure 9, we give the evolution of capacity of underwater communication link using 4-QAM/256-QAM modulation under four absorption seawaters conditions; the four absorption seawater and four signal wavelength are 410 nm ($n_I = 0.3588 \times 10^{-9}$), 470 nm ($n_I = 0.4487 \times 10^{-9}$), 510 nm ($n_I = 1.0549 \times 10^{-9}$) and 570 nm ($n_I = 3.4010 \times 10^{-9}$). Figure 9 reveals that as the absorption coefficients of seawater in the two links are similar, the link with a long signal wavelength has a large capacity, while when the signal wavelengths in the two links are similar, the link with a small absorption coefficient has a large capacity. This result suggest that the carrier wavelength should be adjusted according to the absorptivity of seawater in order to keep the underwater communication link in the optimal state of large capacity.

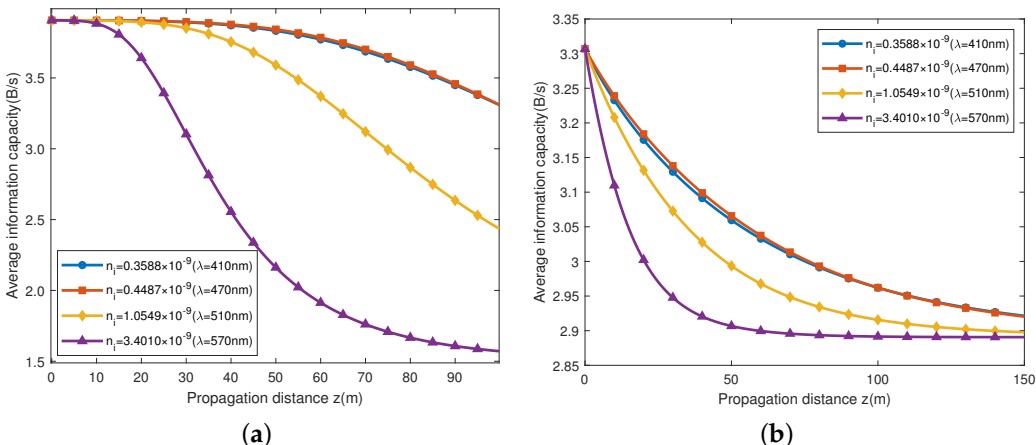

**Figure 9.** The capacity of the POV carrier link versus propagation distance for wavelength and image refractive index under (**a**) 4-QAM, $z = 100$ m, (**b**) 256-QAM, $z = 150$ m.

Figure 10 show the AIC curves of the 4-QAM/256-QAM modulation system for different turbulence strength versus the ratio of temperature and salinity fluctuation. As can be found, the AIC decreases with the increase of temperature and salinity fluctuation. That is, the salinity fluctuation has a greater impact on the AIC than the temperature fluctuation, this conclusion is consistent with the research conclusions of other types of optical vortices [11,12].

The evolution of the AIC of OAM signal carried by POV with the turbulent inner scale and turbulent outer scale of seawater is shown in Figure 11. As can be seen from Figure 11, AIC increases with the increase of turbulent inner scale, but it decreases with the increase of turbulent outer scale. According to the theory of turbulence effect, the turbulent inner scale mainly produces the forward scattering of the emitted beam. Therefore, as the turbulent inner scale increases, the uniform area in the seawater increases, and the wavefront distortion of OAM signal mode passing through the channel decreases; that is, the transmittance of the OAM signal mode increases. On the other hand, the turbulent outer scale mainly leads to the random deflection of the beam propagation path, and the value of the random deflection is directly proportional to the size of the turbulent outer scale. As a result, the transmission channel with large outer scale of turbulence will cause a large random deflection of light rays, resulting in large random optical path difference between the sub-beams of OAM mode, resulting in large wavefront distortion of the OAM mode and large OAM crosstalk. Of course, the AIC loss of the link increases as a result.

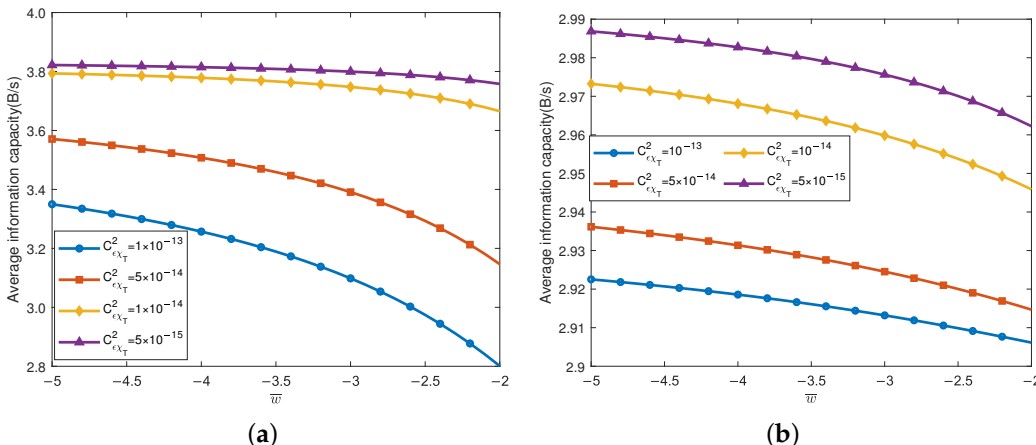

**Figure 10.** The capacity of the POV carrier link versus the ratio of temperature and salinity fluctuation contributions to the refractive index spectrum for the structural constant of the salinity-temperature fluctuation $C^2_{\epsilon\chi_T} = 5 \times 10^{-15}$, $C^2_{\epsilon\chi_T} = 10^{-14}$, $C^2_{\epsilon\chi_T} = 5 \times 10^{-14}$, $C^2_{\epsilon\chi_T} = 10^{-13}$ under (**a**) 4-QAM, $z = 100$ m, (**b**) 256-QAM, $z = 150$ m.

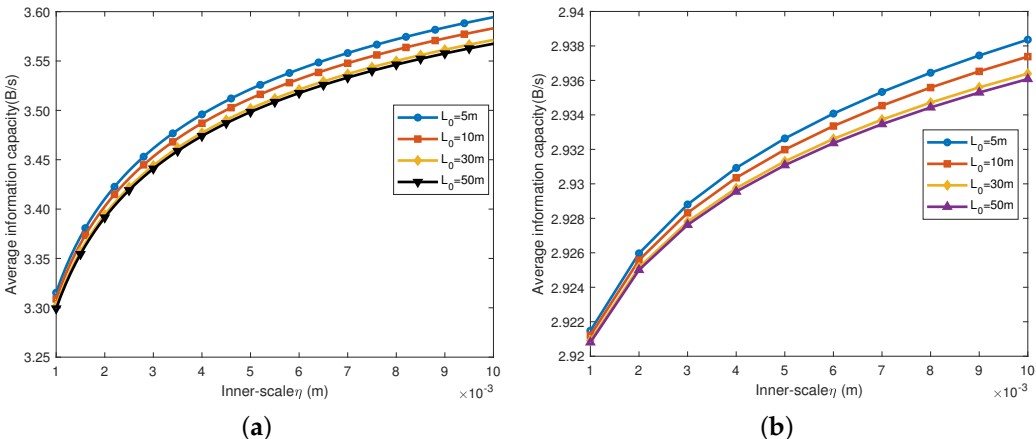

**Figure 11.** The capacity of the POV carrier link versus the inner scale for different outer scale under (**a**) 4-QAM, $z = 100$ m, and (**b**) 256-QAM, $z = 150$ m.

It can be seen from Figure 9–11, the influence of the absorption coefficient of seawater, $\tilde{w}$, the inner scale and outer scale of turbulence on the average information capacity of OAM links with the low-order QAM modulation is greater than that of OAM links modulated by high-order QAM.

## 6. Conclusions

In this paper, we focus on the parameter optimization of POV's source for the underwater transmission system using M-QAM based on bandwidth-limited ABER and study the influence of seawater turbulence and seawater absorption on AIC of bandwidth-constrained communication systems under the optimal parameters of POV's source. The numerical analysis of the ABER and AIC of bandwidth-constrained link is obtained: There exists optimizing half beam-width $\tilde{w}_{opt}(z_0, m_0)$ of POV that is the function of the topological charge $m_0$ and transmission distance. The ABER of bandwidth-constrained link increases with the increase of bandwidth $B$ and the decrease of $P_{TX}/N_0$, that is the increase of signal-to-noise ratio (SNR) of the optical system $P_{TX}/N_0$ enhances the ability of the communication system to resist turbulent channel disturbance. For communication links with short link length, weak seawater turbulence and narrow communication bandwidth, it is more beneficial to adopt QAM of low order $M$. However, for communication systems with long links, strong seawater turbulence and wide communication bandwidth, QAM of high order $M$ should be adopted. High-order modulation is less affected by the turbulent

strength, which indicates that high-order modulation has better anti-interference ability than low-order modulation for long link. Bandwidth-limited ABER increases and AIC decreases with the increase of seawater turbulence strength, and the salinity fluctuation has a greater impact on the AIC than the temperature fluctuation. The decrease of the outer scale and increase of the inner scale as well as channel number have a positive gain for AIC. The results of this paper show that the loss of AIC of underwater transmission system can be reduced by selecting appropriate modulation order of QAM and optimized parameters of POV.

**Author Contributions:** Conceptualization, Y.Z. (Yixin Zhang); methodology, Y.Z. (Yixin Zhang); software, Q.Y.; validation, Y.Z. (Yixin Zhang), Q.Y. and Y.Z. (Yun Zhu); formal analysis, Q.Y.; investigation, Q.Y.; resources, Y.Z. (Yixin Zhang); data curation, Q.Y.; writing—original draft preparation, Y.Z. (Yun Zhu) and L.Y.; writing—review and editing, Y.Z. (Yixin Zhang); visualization, Q.Y.; supervision, Y.Z. (Yixin Zhang) and Y.Z. (Yun Zhu); project administration, Y.Z. (Yixin Zhang) and Y.Z. (Yun Zhu); funding acquisition, Y.Z. (Yixin Zhang) and L.Y. All authors have read and agreed to the published version of the manuscript.

**Funding:** This research was funded by National Natural Science Foundation of China (61871202, 11904136).

**Institutional Review Board Statement:** Not applicable.

**Informed Consent Statement:** Not applicable.

**Data Availability Statement:** Not applicable.

**Conflicts of Interest:** The authors declare no conflict of interest.

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
