# Peer review of "Absorptive Turbulent Seawater and Parameter Optimization of Perfect Optical Vortex for Optical Communication"

_jmse, doi:10.3390/jmse10091256_

Round 1

Reviewer 1 Report (Previous Reviewer 2)

1. Average bit-error probability

2. System model needs to add

3. The paper looks very poorly written 

4. Figure 6 also shows that for  modulation order M ≤ 16 and the SNR PTX/N0 < 25dB , the ABER of link decreases with  the increase of SNR PTX/N0, but when PTX/N0 > 25dB , the increase of PTX/N0 does not improve the link ABER performance., pls explain. 

Author Response

Question:

  1. Average bit-error probability

The authors reply: Thank you very much for your advice. We have changed “Average bit-error probability” into “Average bit-error rate” in the revised manuscript.

  1. System model needs to add

The authors reply: Thank you very much for your advice. We have added the system model description diagram (see Figure 1) to the revised manuscript.

  1. The paper looks very poorly written

The authors reply: Thank you very much for your advice. According to your comment, we have modified the deficiencies in the revised manuscript.

  1. Figure 6 also shows that for modulation order M ≤ 16 and the SNR PTX/N0 < 25dB , the ABER of link decreases with the increase of SNR PTX/N0, but when PTX/N0 > 25dB , the increase of PTX/N0 does not improve the link ABER performance., pls explain.

The authors reply: Thank you very much for your advice. We have added an explanation to this part in the revised manuscript. The explanation is “The reason for this result is that the total noise in ASCNR is the sum of the noise components of the optical system and the turbulence-induced OAM crosstalk noise components [ see Eq. (9)]. When the  is fixed, the noise of the optical system decreases with the increase of  thus the contribution of the noise component of the optical system to the total noise decreases, the turbulence-induced OAM crosstalk noise is the dominant noise. Further from the relationship between ABER and the ASCNR [Eq. (11)] , it is not difficult to understand the numerical results of this graph.”

Reviewer 2 Report (Previous Reviewer 1)

The reviewers are agree to accept by modifying some minor amendments in this manuscript. The authours are advised to read carefully the submitted manuscript and modify accordingly.

1. The reviwers are wondering that the authors didn't provide the gaps between sentenses in multiple places.

2. The authors are advised to define the parameters are used in equation (1-7)in a seperate table.

3.The reviewers asked to modify the sub-heading of each section as the first letter should be in upper case. As an example "5.1. Probability distribution and received probability" should be "Probability Distribution and Received Probability".

Author Response

The reviewers are agree to accept by modifying some minor amendments in this manuscript. The authours are advised to read carefully the submitted manuscript and modify accordingly.

  1. The reviwers are wondering that the authors didn't provide the gaps between sentences in multiple places.

The authors reply: Thank you very much for your advice. We have provided the gaps between sentences in multiple places in the revised manuscript.

  1. The authors are advised to define the parameters are used in equation (1-7)in a seperate table.

The authors reply: Thank you very much for your advice. According to your comment, we have listed the important parameters in Table 1 in the revised manuscript.

3.The reviewers asked to modify the sub-heading of each section as the first letter should be in upper case. As an example "5.1. Probability distribution and received probability" should be "Probability Distribution and Received Probability".

The authors reply: Thank you very much for your advice. According to your comment, we have modified the sub-heading of each section in the revised manuscript.

This manuscript is a resubmission of an earlier submission. The following is a list of the peer review reports and author responses from that submission.

Round 1

Reviewer 1 Report

The manuscript is presented the optimization of the parameters of perfect optical vortex (POV) in underwater wireless optical communication link under M-QAM which is investigated by average bit-error rate (ABER).

However, the reviewers have minor concerns about the presented manuscript.

1.The manuscript should be improve by autors to read carefully. There are plenty of punctuation (gaps, commas) and  indentation errors to present paragraphs and equations. As an example equation (1)-(16) should be comma, colon or full stop in the end.   

2. What is the necessity to optimize the parameters for absorptive turbulece phenomenon for optical communication? The motivation and contributions are missing. It is advise to add a seperate paragraph about the main motivation and related contribution by this research.

3. The plenty of parameters are used in this manuscript. It would be better if authors add a seperate table to define the parameters are used. 

4. Why did the authors are considered absorption? Why not other fading factors such as scattering, bubbles, physio-chemical properties of water,  and pointing error (Misalignment phenomena of the transcivers ).

5. The first letters of subsection (5.1 to 5.4) should be in upper case.

6. The figure 2, 3, 4, and 6 should be large in size as figure 5 or 7 in more readable format.

7. The reviewers suggest to cite the following works in submitted manuscript

https://ieeexplore.ieee.org/document/9707771

https://link.springer.com/article/10.1007/s11831-019-09354-8

Author Response

Dear Editor and reviewer:

Thank you for your letter and comments on our paper entitled " Absorptive turbulent seawater and parameter optimization of perfect optical vortex parameters for optical communication". We appreciate your comments on the shortcomings of the manuscript, and consider these comments to be very valuable, which will help improve the academic level of our manuscript. We have revised the original manuscript (red font) according to your opinions, including changing the title and the content order, simplifying and deleting formulas, etc. The specific modifications are explained as follows:

The manuscript is presented the optimization of the parameters of perfect optical vortex (POV) in underwater wireless optical communication link under M-QAM which is investigated by average bit-error rate (ABER). However, the reviewers have minor concerns about the presented manuscript.

Question:

  1. The manuscript should be improved by authors to read carefully. There are plenty of punctuation (gaps, commas) and indentation errors to present paragraphs and equations. As an example equation (1)-(16) should be comma, colon or full stop in the end.

The authors reply: Thank you very much for your advice. We have modified the punctuation (gaps, commas) and indentation errors to present paragraphs and equations in the revised manuscript.

  1. What is the necessity to optimize the parameters for absorptive turbulence phenomenon for optical communication? The motivation and contributions are missing. It is advised to add a seperate paragraph about the main motivation and related contribution by this research.

The authors reply: Thank you very much for your advice. We have added motivation to the introduction of the revised draft. The added paragraph is “In the case of selected beam types, selecting appropriate system parameters is also one of the important methods to effectively control the impact of ocean turbulence [24], but the parameter optimization of existing POV has not been studied [15-23].”.

  1. The plenty of parameters are used in this manuscript. It would be better if authors add a seperate

table to define the parameters are used.

The authors reply: Thank you very much for your advice. According to your comment, we have listed the important parameters in Table 1 in the revised manuscript.

  1. (1)Why did the authors are considered absorption? (2) Why not other fading factors such as scattering, bubbles, physio-chemical properties of water, and pointing error (Misalignment phenomena of the transcivers ).

The authors reply: Thank you very much for your advice.

(1) Because even in clean seawater without bubbles and organic solutes, seawater is a strong light absorber, and seawater absorption severely limits the distance that signals can travel in seawater, so absorption must be considered.

(2) Seawater scattering, bubbles, physio-chemical properties of seawater, and pointing error are other important factors affecting optical communication in seawater, but seawater scattering, bubbles and physio-chemical properties of seawater are very complex factors affecting optical communication transmission. When the sea breeze, the physio-chemical properties of the water (organic solute), seawater plants, and fish activity is different, such as scattering caused by sea, bubbles, physio-chemical properties of seawater and pointing error is different, each factor is a need to study the complexity of the subject, so we only consider a POV in clean or ignore the complicated factors optical communication problems in offshore underwater channel. Therefore, this paper considers the absorption and turbulence of seawater. Only the theoretical model based on seawater absorption can describe the law of light transmission in seawater more accurately.

  1. The first letters of subsection (5.1 to 5.4) should be in upper case.

The authors reply: Thank you very much for your advice. According to your comment, the first letters of subsection (5.1 to 5.4) have been revised to be in upper case in the revised manuscript.

  1. The figure 2, 3, 4, and 6 should be large in size as figure 5 or 7 in more readable format.

The authors reply: Thank you very much for your advice. According to your comment, we have modified figures 2, 3, 4, and 6 to the same large in size as figure 5 in the revised manuscript.

  1. The reviewers suggest to cite the following works in submitted manuscript https://ieeexplore.ieee.org/document/9707771 https://link.springer.com/article/10.1007/s11831-019-09354-8

The authors reply: Thank you very much for your advice. According to this comment, we have added these important references in the introduction of the revised manuscript.(Please refer to line68-line76)

Referances

  1. A. S. Ostrovsky, C. Ricrenstorff-Parrao, and V. Arrizón, “Generation of the ‘perfect’ optical vortex using a liquid-crystal spatial light modulator,” Opt. Lett. 38(4), 534–536 (2013).
  2. P. Vaity and L. Rusch, “Perfect vortex beam: Fourier transformation of a Bessel beam,” Opt. Lett. 40(4), 597–600 (2015).
  3. F. Q. Zhu, S. J. Huang, W. Shao, J. Zhang, M. S. Chen, W. B. Zhang, and J. Z. Zeng, “Free-space optical communication link using perfect vortex beams carrying orbital angular momentum (OAM),” Opt. Commun. 396(1), 50–57 (2017).
  4. W. Shao, S. J. Huang, X. P. Liu, and M. S. Chen, “Free-space optical communication with perfect optical vortex beams multiplexing,” Opt. Commun. 427(15), 545–550 (2018).
  5. C. Y. Yang, Y. Lan, X. Y. Jiang, H. Long, J. Hou, and S. P. Chen, “Beam-holding property analysis of the perfect optical vortex beam transmitting in atmospheric turbulence,” Opt. Commun. 472(1), 125879-1- 125879-6 (2020).
  6. M. K. Karahroudi, S. A. Moosavi, A. Mobashery, B. Parmoon, and H. Sghhafifar, “Performance evaluation of perfect optical vortices transmission in an underwater optical communication system,” Appl. Opt. 57(30), 9148-9154 (2018).
  7. Z. Hu, H. Liu, J. Xia, A. He, H. Li, Z. Du, T. Chen, Z. Li, and Y. Lu, “Propagation characteristics of the perfect vortex beam in anisotropic oceanic turbulence,” Appl. Opt. 59(32), 9956-9962 (2020).
  8. W. Wang, P. Wang, W. Pang, Y. Pan, Y. Nie, and L. Guo, “Evolution properties and spatial-mode UWOC performances of the perfect vortex beam subject to oceanic turbulence,” IEEE Transactions on Commun. 69(11), 7647–7658 (2021).
  9. H. Yang, Q. Yan, Y. Zhang, and L. Hu, “Received probability of perfect optical vortex in absorbent and weak turbulent seawater links,” Appl. Opt. 60(35), 10772-1079 (2021).
  10. H. Yang, Q. Yan, P. Wang, and L. Hu, Y. Zhang, “Bit-error rate and average capacity of an absorbent and turbulent underwater wireless communication link with perfect Laguerre-Gauss beam,” Opt. Express 30(6), 9050–9064 (2022).

Reviewer 2 Report

1. Without having a system model equation, it is hard to project what is this about for readers. 

2. You need to remove many introductory sentences from the article. As this is a scientific paper, we can assume readers know some details. 

 (14) The log function from the above equation relates to the base 2. Therefore, the mutual 140 information represents the amount of information (per symbol) that is conveyed by the 141 channel, which represents the uncertainty about the channel input that is resolved by 142 observing the channel output. The mutual information, i.e., the information conveyed by 143 the channel, is obtained as the output information minus information lost in the cha

3. A few important references are missing 

1.  "Recent Advances and Future Directions on Underwater Wireless Communications",  Archives of Computational Methods in Engineering, Aug 2019. https://doi.org/10.1007/s11831-019-09354-8 

2. “Asymmetric Satellite-Underwater Visible Light Communication System for Oceanic Monitoring,” IEEE Access,  vol. 7,  pp. 133342 - 133350, Aug 2019. 

3. “O2O: An Underwater VLC Approach in Baltic and North Sea”, Electronics, Jan 2022. 

4.  “Recent Trends in Underwater Visible Light Communication (UVLC) Systems”, IEEE Access, Feb 2022.  

5. "Secure NOMA Assisted Multi-LED Underwater Visible Light Communication", IEEE Transactions on Vehicular Technology,  10.1109/TVT.2022.3167992

4. Contribution is not clear 

5. Explain equation 15 and 16 carefully. It is good to take more space to explain equation and figures than introductory items. 

6. The conclusion is too lengthy, pls move some parts to the body

7. Add explanation about potential application about this. 

Author Response

(The authors gave the same response as above.)

Reviewer 3 Report

Please reconsider the selection of the propagation environment parameters. For example: When light propagates 150 m in a real ocean, it will loose so much of the intensity that the detection would be nearly impossible. More examples could be easily found in your work. In order to provide useful insight to the community, it is an imperative that you perform the numerical analysis that reflects real propagation medium.

Author Response

Dear Editor and reviewer:

Thank you for your letter and comments on our paper entitled " Absorptive turbulent seawater and parameter optimization of perfect optical vortex parameters for optical communication". We appreciate your comments on the shortcomings of the manuscript, and consider these comments to be very valuable, which will help improve the academic level of our manuscript. We have revised the original manuscript (red font) according to your opinions, including changing the title and the content order, simplifying and deleting formulas, etc. The specific modifications are explained as follows:

Question:

Please reconsider the selection of the propagation environment parameters. For example: When light propagates 150 m in a real ocean, it will loose so much of the intensity that the detection would be nearly impossible. More examples could be easily found in your work. In order to provide useful insight to the community, it is an imperative that you perform the numerical analysis that reflects real propagation medium.

The authors reply:

Thank you very much for your comment. As the transmission distance of 150m is the distance that the experiment can reach at present [40,41], due to ocean turbulence, absorption and various losses, the signal loss at 150m is relatively large. The received probability of transmission distance of 50m and 100m is shown below, and it can be found that the longer the distance, the lower the probability of received signal.

  • (a)
  • (b)

The received probability of OAM signal modes carried by POV versus different topological charge of initial OAM modes and topological charge of new OAM modes for r0=0.03m, ϖ=-4.5, nI=0.4487×10-9.  (a) z=50m  (b) z=100m

  1. S. Zhu, X. Chen, X. Liu, G. Zhang, P. Tian, “Recent progress in and perspectives of underwater wireless optical communication,” Progress in Quantum Electronics 73, 100274, (2020).
  2. X. Chen, X. Yang, Z. Tong, Y. Dai, X. Li, M. Zhao, Z. Zhang, J. Zhao, and J. Xu, “150 m/500 Mbps Underwater Wireless Optical Communication Enabled by Sensitive Detection and the Combination of Receiver-Side Partial Response Shaping and TCM Technology,” J. Lightwave Technol. 39(14), 4614-4621 (2021).
